# THE BRAIN'S BITTER LESSON: SCALING SPEECH DECODING WITH SELF-SUPERVISED LEARNING

## ABSTRACT

The past few years have produced a series of spectacular advances in the decoding of speech from brain activity. The engine of these advances has been the acquisition of labelled data, with increasingly large datasets acquired from single subjects. However, participants exhibit individual differences, such as anatomy, and datasets use varied scanners and task designs. As a result, prior work has struggled to leverage data from multiple subjects, multiple datasets, multiple tasks, and unlabelled datasets. In turn, the field has not benefited from the rapidly growing number of open neural data repositories to exploit large-scale data and deep learning. This gap exists for all neural data, but especially for magnetoencephalography (MEG), where the scale of individual datasets has not yet caught up with other modalities. To address this, we develop a set of neuroscience-inspired self-supervised objectives, together with a neural architecture, for representation learning from heterogeneous and unlabelled neural recordings. Experimental results with MEG show that representations learned with these objectives scale with data, generalise across subjects, datasets, and tasks, outperform using the raw input representation, and even surpass comparable self-supervised approaches. In addition, we set new benchmarks for two foundational speech decoding tasks. Collectively, these methods now unlock the potential for training speech decoding models with orders of magnitude more existing data.

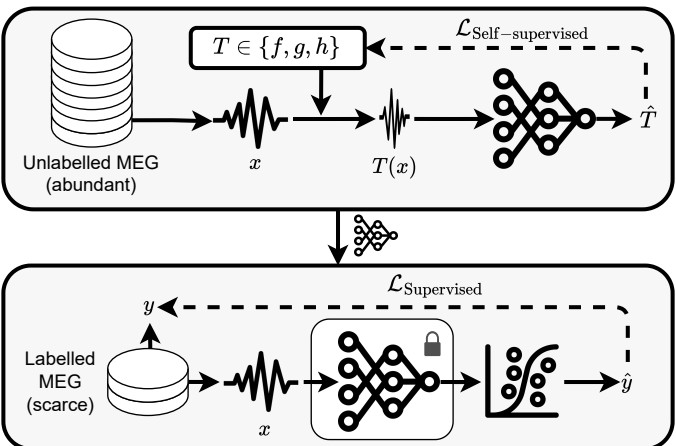

Figure 1: **Leveraging unlabelled data using pretext tasks for speech decoding.** We pre-train a neural network using tasks that generate implicit labels from abundant unlabelled MEG neuroimaging data, permitting learning from large heterogeneous datasets. The tasks apply a randomly selected neuroscientifically relevant transformation $T$ to the data and the network predicts the transformation. We then train a linear probe on top of the pre-trained model, which remains frozen, with labelled data, achieving superior generalisation (cf. raw inputs) owing to the strength of the representation.

# 1 INTRODUCTION

In his *Bitter Lesson*, Richard Sutton argues that a major conclusion of 70 years of AI research is that general methods exploiting large-scale computation will outperform model-based approaches as the availability of compute increases (Sutton, 2019). In line with this, the generality of deep learning, via statistical learning from ever bigger datasets, has allowed the field to leverage computation in a way that appears to scale arbitrarily, leading to astounding advances across a diverse set of domains (Jumper et al., 2021; Caron et al., 2021; OpenAI, 2023; Radford et al., 2023).

In the domain of brain data, and of tasks like speech decoding, the bitter lesson has not yet been fully assimilated. State-of-the-art *brain-computer interfaces (BCIs)* have tried to scale up labelled datasets for individual subjects, using either invasive (Moses et al., 2021; Willett et al., 2023) or non-invasive brain recordings (Tang et al., 2023), mapping these to transcripts of attempted or imagined speech. Yet, a number of obstacles to scale remain. With few exceptions at present, e.g. Défossez et al. (2023), speech decoding models tend not to train on data from more than one subject. Moreover, they do not combine data from multiple datasets and in general do not utilise unlabelled data, or data from diverse tasks. Thus the size of training data has been limited to how much can be acquired for a single subject, and data from other subjects, or from the growing number of public data repositories, has not been leveraged. There are many reasons for these limitations; individual brains and data from different neuroimaging scanners differ, for example. But overcoming these limitations, as has begun to happen in neighbouring sub-fields, such as Jiang et al. (2024), holds the promise of training models on collective, internet-scale data.

While neuroimaging modalities such as electroencephalography (EEG) are more abundant, MEG may be a better modality for decoding as it provides a richer signal (Lopes da Silva, 2013; Hall et al., 2014). Given the scarcity of speech-labelled MEG data and the relative abundance of other MEG data, *self-supervised learning (SSL)* appears promising as it is an avenue for domains where labels are rare or hard to obtain (Balestriero et al., 2023). But the scale of public MEG data, while large, is still not at the volume of breakthroughs in self-supervised image and natural language processing, let alone EEG. Thus, SSL methods for MEG need to be highly data-efficient. *Pretext* tasks are one such method in which domain-specific self-supervised tasks are used to pre-train a model on unlabelled data by generating implicit training labels through transformations of the input in order to help a downstream task. We develop a set of these tasks, informed by advances in neuroscience, for learning with unlabelled brain data (Figure 1) and design an architecture for processing continuous multi-sensor neuroimaging signals which we train using our pretext tasks. In order to scale existing non-invasive datasets, we provide a unified method that allows us to leverage data from other experiments that do not have the same labels (by treating them as unlabelled) and that come from different subjects and neuroimaging scanners. We evaluate the representations learned with our approach on heard speech datasets acquired with non-invasive MEG, setting the baselines for speech detection and voicing classification on this data. The results not only demonstrate that scaling with unlabelled data works in speech decoding, but also shows that these representations can generalise across datasets, tasks, and even novel subjects for the first time. Our main contributions are:

- A set of domain-specific **self-supervised pretext tasks** for representation learning that can scale speech decoding over multiple subjects, multiple studies, and unlabelled data;

- A data-efficient **neural architecture** for learning these self-supervised objectives and training downstream speech decoding from brain data; and

- A comprehensive **experimental evaluation**, using multiple times the volume of data in prior work, that verifies the above claims and additionally provides evidence for the existence of **scaling laws** when pre-training models with unlabelled MEG recordings.

# 2 RELATED WORK

Prior work in speech decoding has focused almost entirely on supervised learning with decoding models that typically do not generalise across participants or experiments. This is true both in recent state-of-the-art invasive studies (Moses et al., 2021; Metzger et al., 2023; Willett et al., 2023; Chen et al., 2024a) and non-invasive studies (Tang et al., 2023). These prior works have scaled up the experimental data collected within individual subjects, but are unable to leverage data from

other subjects and experiments. Focusing on semantic rather than phonetic decoding, the method developed by Tang et al. (2023) is remarkable for showing an ability to generalise across labelled task data when listening to speech, imagining speech, or even watching videos. They do not, however, leverage unlabelled data and are unable to show generalisation between subjects.

Specific studies into the limitations of generalising models between subjects show that while performance decreases on average when subjects are pooled, there are exceptions (e.g. Anumanchipalli et al. (2019) and Makin et al. (2019) in surgical settings and Csaky et al. (2022) non-invasively). Exploiting audio data in a multi-modal framework, Défossez et al. (2023) show that decoding performance improves for a segment identification task as data from multiple subjects listening to connected speech are aggregated. However, they do not demonstrate the ability to generalise to novel subjects and must retrain their model for new datasets. Moreover, although they repeat the result within two MEG and two EEG datasets, Défossez et al. (2023) do not show any improvements for pooling data across datasets. Their method is also unable to incorporate data without corresponding audio labels and so they do not combine data from studies with other kinds of labels either; cf. Wang & Ji (2022); Duan et al. (2023); Wang et al. (2023a). Unfortunately, the first two of these papers included a bug in their evaluation code. As such, their methods may perform no better than a baseline that provides pure noise inputs to the model (Jo et al., 2024).

In general, speech decoding has centred on different kinds of speech: listening, imagining, speaking out loud, and, for paralysed patients, attempting to speak aloud. We focus here on listening because it is easier to decode than imagined speech (e.g. Martin et al. (2014)). There is also some evidence of a functional overlap between listening and imagined speech representations in the brain (Wandelt et al., 2024), though we acknowledge that the question of overlap has been contested (Langland-Hassan & Vicente, 2018). Prior work has also investigated the two tasks that we focus on here (Dash et al., 2020; Moses et al., 2021; Gwilliams et al., 2023). The first of these, speech detection, formed the backbone to Moses et al. (2021), where a speech detection model was trained and subsequently used to detect isolated words, which were in turn classified and checked against a language model to generate acceptable sentences. Hamilton et al. (2018) further elaborated on the neural anatomy underlying speech detection, categorising neural responses in the *superior temporal gyrus (STG)* to sustained speech and speech onset. As for the second task, voicing classification, Gwilliams et al. (2023) used this task as a proxy for phoneme classification, as pooling phonemes into unvoiced or voiced segments (e.g. /p t k f s/ vs /b d g v z/) improves data efficiency. We note that voicing classification and speech detection are related tasks as voicing is a subclass of speech. This makes them foundational for building hierarchical speech decoding pipelines similar to prior surgical decoding work (Moses et al., 2021; Willett et al., 2023).

In the computer vision literature, there have been a plethora of methods that use self-supervised pretext tasks for representation learning (Agrawal et al., 2015; Doersch et al., 2015; Noroozi & Favaro, 2016; Larsson et al., 2016; Zhang et al., 2016; Gidaris et al., 2018). Until now, similar approaches have not translated to the brain decoding literature with few exceptions (e.g. Cai et al. (2023)). However, prior work has used other methods to leverage unlabelled brain data (Banville et al., 2019; Kostas et al., 2021; Le & Shlizerman, 2022; Zhang et al., 2023; Yi et al., 2023; Ye et al., 2023; Yuan et al., 2024; Chen et al., 2024b). For example, Jiang et al. (2024) succeeded in cross-dataset and cross-task generalisation, using a transformer with tokenised brain signals and a masked token prediction objective. Although this work combined unlabelled datasets, their results studied simpler non-speech tasks with EEG. Wang et al. (2023b) used a similar approach, replacing tokens with contextualised embeddings of time-frequency input representations. Their impressive speech detection results were achieved with invasive neural recordings, which are comparatively rare and thus have much less potential to scale than non-invasive data. Perhaps the closest work to ours in terms of unlocking scaling with neural data is BIOT (Yang et al., 2023). This is a self-supervised architecture for encoding bio-signals that is similarly capable of training with different datasets, labels, and varied numbers of sensors. Like the previous works, the approach tokenises signals for a transformer architecture, but instead of a masked loss it uses a contrastive pre-training objective. While theoretically supporting MEG, Yang et al. (2023) evaluate BIOT on simple ECG/EEG tasks rather than address the comparatively complex challenge of speech decoding with MEG data.

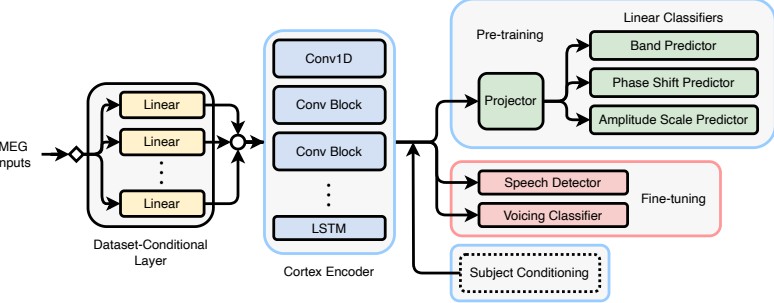

Figure 2: **Architecture overview.** Inputs are projected into a shared dimension by the dataset-conditional layer, then encoded. In pre-training, all weights are trainable except for modules in light-red, while in fine-tuning, modules with light-blue borders are frozen and modules with light-red borders are unfrozen. Dashed borders indicate optional components.

## 3 METHOD

To encode continuous neuroimaging data, we introduce a neural architecture to embed heterogeneous brain signals. We leverage this architecture for self-supervised learning from unlabelled MEG data using a set of pretext tasks designed to generate generalisable brain representations for speech decoding. With this approach, we hope to replicate similar successes in computer vision (Gidaris et al., 2018; Chen et al., 2020).

### 3.1 NETWORK ARCHITECTURE

Our two-stage neural network architecture (Figure 2) uses pretext tasks in pre-training to learn a representation with unlabelled brain data. Then, the fine-tuning stage uses this representation to learn the downstream task by training with labelled data.

We divide recordings into windows of length $w$ seconds or $t$ samples. At train time, each batch of windows is standardised such that each sensor has zero mean and unit variance. The network takes as input the standardised sample windows. To combine heterogeneous datasets, which have different numbers of sensors $S$, we apply a dataset-conditional linear layer to the sensor dimension, projecting the signal into a shared space with dimension $d_{\text{shared}}$. Then, to encode the signal, we construct a wave-to-wave convolutional encoder architecture, the *cortex encoder*, inspired by work in neural audio codecs (Zeghidour et al., 2022; Défossez et al., 2022). Specifically, our convolutional encoder adapts the implementation of the SEANet architecture (Tagliasacchi et al., 2020) used in Défossez et al. (2022) which we describe here and as part of Figure 2. As these codecs typically operate on mono audio signals in $\mathbb{R}^{1 \times t}$, while our signals are in $\mathbb{R}^{d_{\text{shared}} \times t}$, we increase the convolutional channel dimension from 1 to match $d_{\text{shared}}$ while also inflating the channel dimension of subsequent convolutions. We refer to the output dimension of embeddings from this backbone as $d_{\text{backbone}}$. Thus, the backbone takes as input a window in $\mathbb{R}^{S \times t}$, and encodes this into $\tau$ embeddings (where $\tau < t$), each of dimension $d_{\text{backbone}}$ (i.e. an $\mathbb{R}^{d_{\text{backbone}} \times \tau}$ output).

Just as speakers have different voices, neural responses between subjects have different characteristics. Consequently, individual variation leads to models that do not generalise well across subjects (Csaky et al., 2022). In the speech literature, models include speaker conditioning to account for these differences (Gibiansky et al., 2017). We take a similar approach by introducing subject conditioning. Zeghidour et al. (2022) find that conditioning is equally effective at the encoder bottleneck as in other stages of the model. Hence, we place ours at the cortex encoder bottleneck for simplicity. We use *feature-wise linear modulation (FiLM)* (Perez et al., 2018) as our conditioning method.

Following the advice of Balestriero et al. (2023, Section 3.2), we use a two-layer feedforward projector to alleviate misalignment between our pretext and downstream tasks in the representation. After the projector, linear classifiers make predictions for each of the pretext tasks. When fine-tuning, we train a linear decoder, for a downstream task, on top of the pre-trained representation, which remains

frozen. Thus, we backpropagate only through the classifier. A trainable dataset-specific linear layer can be introduced for a novel dataset.

For speech detection, our classifier makes a prediction for each individual embedding. For voicing classification, where there is only one label for each sample window, the embeddings are flattened into a tensor in $\mathbb{R}^{d_{\text{backbone}} \times \tau}$ representing the entire window. This is the input to the voicing classifier and is referred to as full epoch decoding in neuroimaging literature (Csaky et al., 2023).

## 3.2 PRETEXT TASKS

Our pretext tasks are unsupervised feature learning tasks that aim to learn generalisable speech decoding features. Since different datasets use varied numbers of sensors, we construct these tasks with labels that are agnostic to the number of sensors in the signal.

**Band prediction.** In the literature, neural responses can be segmented into functional frequency bands (Giraud & Poeppel, 2012; Piai et al., 2014; Mai et al., 2016). *Delta (δ)* waves (0.1–4 Hz) are commonly associated with the rhythmic structure of heard speech (Luo et al., 2010), *Theta (θ)* waves (4–8 Hz) reliably track (Luo & Poeppel, 2007) and phase-lock to the amplitude envelope of heard sentences (Peelle et al., 2012), *Alpha (α)* waves (8–12 Hz) relate to attentional processes and the inhibition of irrelevant information, helping to focus on relevant speech signals (Strauß et al., 2015), *Beta (β)* waves (12–30Hz) are implicated in top-down predictive coding (Bressler & Richter, 2015) which affects lexical processing (Weiss & Mueller, 2012), *Gamma (γ)* waves (30–70 Hz) occur with higher cognitive functions (e.g. memory, learning, reasoning, and planning) (Fries, 2009; Buzsáki & Wang, 2012), and *High Gamma ($\gamma^{\text{high}}$)* waves (>70 Hz) have been linked specifically to speech detection (Hamilton et al., 2018) and phonemic feature classification in the STG (Mesgarani et al., 2014) as well as phonemic feature classification in the *ventral sensorimotor cortex (vSMC)* (Cheung et al., 2016). As High Gamma is a relatively wide band, we have split it into two sub-bands: *Lower High Gamma ($\gamma^{\text{high}}_{\text{lower}}$)* waves (70–100 Hz) and *Upper High Gamma ($\gamma^{\text{high}}_{\text{upper}}$)* waves (100–150 Hz).

To learn representations that can distinguish between these, our band prediction task applies a band-stop filter for a randomly selected band $\omega$ to the sample $x$, passes the filtered sample $x^{\omega'}$ through the network backbone $g$ and the corresponding linear predictor $f_{\text{band}}$, requiring the network to classify which frequency band $\omega$ was rejected. This yields the loss

$$\mathcal{L}_{\text{band}} = \sum_{x \in B} \mathcal{L}_{\text{CE}}(f_{\text{band}}(g(x^{\omega'})), \omega), \tag{1}$$

where $B$ is a mini-batch of samples, $\omega \in \{\delta, \theta, \alpha, \beta, \gamma, \gamma^{\text{high}}_{\text{lower}}, \gamma^{\text{high}}_{\text{upper}}\}$, and $\mathcal{L}_{\text{CE}}$ is the cross-entropy loss as this is a multi-class classification task.

**Phase shift prediction.** Phase coupling between networks of neuron populations is necessary for coordinating brain activity (Fries, 2005; Vidaurre et al., 2018). Thus, since phase often synchronises between communicating brain areas, phase coupling between spatially distant sensors is likely to be a useful feature. Supporting this insight, recent work (Jiang et al., 2024) also finds phase to be an essential component of the signal.

To learn representations that encode phase differences between brain areas, this task applies a discrete uniform random phase shift $\phi \in \{0, \frac{\pi}{8}, \frac{\pi}{4}, \frac{3\pi}{8}, \frac{\pi}{2}, \frac{5\pi}{8}, \frac{3\pi}{4}, \frac{7\pi}{8}\}$ to a uniformly randomly selected proportion $\rho$ of the sensors. Applying this shift to random sensors is critical since sensors are placed in different positions, capturing different regions of the brain. Uniform random selection ensures differences between any two regions of the brain are represented. The objective of this task is to predict the phase shift. This leads to a similar loss

$$\mathcal{L}_{\text{phase}} = \sum_{x \in B} \mathcal{L}_{\text{CE}}(f_{\text{phase}}(g(x^{\phi})), \phi), \tag{2}$$

where $x^{\phi}$ describes the signal with a phase shift $\phi$ applied to a proportion of the sensors. We use a discrete number of possible phase shifts, treating it as a multi-class task rather than a regression task, to ease the difficulty of the problem as MEG scanners typically have a large number of sensors.

**Amplitude scale prediction.** MEG and EEG signals use an array of sensors at different spatial locations, capturing different signal sources more intensely. Representing the relative amplitude dif-

ference between sensors could be important for differentiating between neural responses originating from distinct parts of the brain. Within speech, Hamilton et al. (2018) find that localised regions of the STG respond to sustained speech and speech onsets. Differentiating between neural responses from this region and others may be essential for decoding speech perception.

Thus, this pretext task focuses on learning representations that encode relative sensor amplitude differences. Similar to the phase shift task, we select a random proportion of the sensors $\rho$ and apply a discrete random amplitude scaling coefficient $A \in [-2, 2]$, discretised into 16 scaling factors, to the signal. The objective is to predict the scaling factor, leading to the loss

$$\mathcal{L}_{\text{amplitude}} = \sum_{x \in B} \mathcal{L}_{\text{CE}}(f_{\text{amplitude}}(g(x^A)), A), \tag{3}$$

where $x^A$ is the signal scaled with $A$.

These pretext tasks capture complementary time- and frequency-domain properties of the signal. Hence, during pre-training, we combine them, creating an augmented version of the input for *every* pretext task by applying the matching transformation. We feed the augmented inputs through the network backbone and apply the corresponding classifier to predict the transformation, summing the weighted losses such that our final pre-training loss is given by

$$\mathcal{L}_{\text{SSL}} = w_1 \mathcal{L}_{\text{band}} + w_2 \mathcal{L}_{\text{phase}} + w_3 \mathcal{L}_{\text{amplitude}}, \tag{4}$$

where $w_i$ is a constant coefficient for each self-supervised loss.

## 4 EXPERIMENTS

In this section, we evaluate the representations learned with our pretext tasks by measuring their ability to scale downstream performance with unlabelled data. This includes understanding how well they can generalise across datasets, subjects, and tasks. We focus our evaluation on MEG data as the signal is rich, with better spatial resolution than EEG (Lopes da Silva, 2013) and faster sampling rates than fMRI (Hall et al., 2014).

We pre-train all models to completion and then fine-tune on labelled data for each task. In all tables and figures, we quote the *receiver operating characteristic area under the curve (ROC AUC)* where chance is always 0.5 regardless of class balance. We show the test ROC AUC at the best validation ROC AUC (early stopping) and quote uncertainty as the standard error of the mean over three seeds. Additionally, we state the $t$-score and $p$-value from single-sample one-sided $t$-tests against chance.

### 4.1 EXPERIMENTAL SETUP

**Datasets.** Unless specified otherwise, our experiments use Cam-CAN (Shafto et al., 2014; Taylor et al., 2017) as an unlabelled representation learning dataset for pre-training. This is a study containing 641 subjects with resting and sensorimotor tasks, totalling approximately 160 hours of MEG recordings. For our downstream tasks, we use two labelled heard speech MEG datasets where participants listen to short stories or audiobooks. Armeni et al. (2022) contains 3 subjects who listen to 10 hours of recordings each (30 hours total) while Gwilliams et al. (2023) has 27 subjects, each recorded for 2 hours (54 hours total). Overall, we utilise over 200 hours of data. To the best of our knowledge, this is the largest volume of MEG data ever used for speech decoding.

**Preprocessing.** Each recording is in $\mathbb{R}^{S \times T}$ where $S$ is the number of sensors and $T$ is the number of time points sampled by the scanner. To eliminate high-frequency muscle movement artifacts, we apply a low-pass filter at 125Hz as well as a high-pass filter at 0.5Hz to remove slow-drift artifacts. Since the datasets were recorded in Europe, where the electric grid frequency is 50Hz, we apply a notch filter at multiples of 50Hz to account for line noise. Treating the low-pass filter threshold as the Nyquist frequency, we downsample the signal to twice that at 250Hz, avoiding aliasing within our band of interest. Finally, we detect bad sensor channels, those with significant noise and artifacts, using a variance threshold and replace them by interpolating the spatially nearest sensors.

**Downstream tasks.** We evaluate our methods with two fundamental speech decoding tasks of increasing difficulty. The first, *speech detection*, determines whether speech occurs in the auditory

Table 1: **Pre-training with pretext tasks leads to better representations for speech detection.** In the *linear*-only case, we train a supervised linear classifier on the input MEG signals. For BIOT, we train a linear layer on top of a backbone pre-trained on CamCAN, with the rest of the model frozen. Similarly, for *ours*, we train a linear probe on top of our pre-trained backbone with its weights frozen. In the *no pre-training* baseline, the backbone uses randomly initialised and subsequently unmodified weights. When *all* pretext tasks are used, their losses are weighted equally.

| Experiment | | Armeni | | | Gwilliams | | |
|---|---|---|---|---|---|---|---|
| | | ROC AUC | $t$ | $p$ | ROC AUC | $t$ | $p$ |
| Linear | | $0.559 \pm 2e{-}4$ | 341 | $4e{-}6$ | $0.527 \pm 7e{-}5$ | 379 | $3e{-}6$ |
| BIOT + linear | | $0.500 \pm 4e{-}4$ | 0 | $6e{-}1$ | $0.499 \pm 2e{-}4$ | $-3$ | $1e{+}0$ |
| **Ours** | No pre-training | $0.519 \pm 0.002$ | 8 | $7e{-}3$ | $0.498 \pm 0.003$ | 0 | $7e{-}1$ |
| + linear | $\text{Amp}_{(\rho = 0.2)}$ | $0.602 \pm 0.001$ | 114 | $4e{-}5$ | $0.532 \pm 0.005$ | 6 | $1e{-}2$ |
| | $\text{Phase}_{(\rho = 0.5)}$ | $0.603 \pm 0.003$ | 35 | $4e{-}4$ | $0.535 \pm 0.003$ | 12 | $3e{-}3$ |
| | Band | $0.616 \pm 0.003$ | 44 | $3e{-}4$ | $0.542 \pm 0.001$ | 46 | $2e{-}4$ |
| | All tasks | $\mathbf{0.621} \pm 0.003$ | 36 | $4e{-}4$ | $\mathbf{0.543} \pm 0.003$ | 13 | $3e{-}3$ |

stimulus using the neural response. The second task is *voicing classification*. Given data aligned at the occurrence of a phoneme, the task is to recognise whether the phoneme is *voiced* or *voiceless*, where voicing is a binary phonetic feature that categorises whether a speech sound is associated with vocal cord vibration. We select these tasks as they are simpler than phoneme recognition, but are foundational because they must be solved to decode speech accurately into natural language.

## 4.2 Learning Generalisable Representations Using Pretext Tasks

Our first experiment investigates whether our self-supervised objectives produce generalisable representations. In Table 1, we show the results of pre-training models with each pretext task independently as well as together. Here, all of our pretext tasks lead to results that are statistically significant, and outperform a baseline fine-tuned without pre-training. This provides initial evidence that our tasks are helpful in speech decoding. Interestingly, the combination of all pretext tasks leads to better generalisation than any task on its own. As we hypothesised earlier, this may be because our pretext tasks capture complementary properties in time- and frequency-space, enforcing that our representation includes more salient features for speech decoding than any individual task.

Now, we turn to the other baselines. Our approach significantly outperforms the equivalent with a raw MEG input instead of a pre-trained representation (the *linear* experiment). Here, the baseline has substantially more trainable parameters because the input dimension is far larger without an encoder. Even with this bias favouring the experiment with the raw input, using our representation still performs better. We also compare our approach to BIOT (Yang et al., 2023) which is a similar state-of-the-art self-supervised method. When BIOT is pre-trained using exactly the same data, the fine-tuned probe fails to generalise entirely after exhaustive hyperparameter tuning. We put this down to three critical reasons. Firstly, BIOT was designed around considerably lower-dimensional signals. Their EEG evaluation used an order of magnitude fewer sensors than our MEG data. With MEG, their transformer approach requires many more channel embeddings, leading to difficulty learning the complex interactions between sensors. Secondly, our self-supervised objective extracts speech decoding features which is essential for solving speech decoding tasks. BIOT performs well on simple EEG tasks in Yang et al. (2023)'s evaluation, but non-invasive speech decoding is significantly more challenging. Together, these obstacles suggest a vast amount of data is required to learn their objective with MEG. Indeed, given that they pre-train with over 50 thousand hours of EEG data in their evaluation, their objective appears too general to efficiently learn a representation for speech decoding from the limited amount of MEG pre-training data (160 hours) available to us. This highlights the importance of data-efficiency in SSL methods for MEG.

Among the individual pretext tasks, band prediction leads the rest. Perhaps this is because, by learning to discriminate between meaningful bands, the representation easily identifies phase-locking to speech onset in theta waves (Peelle et al., 2012). Further investigation is necessary here. The choice of the proportion of sensors to apply transformations to, $\rho = 0.5$ for phase shift prediction and

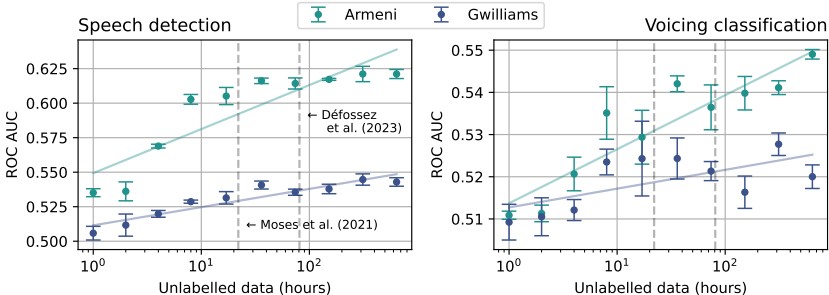

Figure 3: **Scaling unlabelled data improves generalisation.** We pre-train the model on increasing amounts of unlabelled data from Cam-CAN (Shafto et al., 2014; Taylor et al., 2017). The solid lines are the best linear fits to the data and the dashed lines show the amount of data used in prior surgical (Moses et al., 2021) and non-invasive (Défossez et al., 2023) work.

$\rho = 0.2$ for amplitude prediction, were determined through a hyperparameter search. We conjecture that a smaller $\rho$ is optimal for amplitude scale prediction since this leads to representations that are especially strong at discriminating amplitude differences among small groups of sensors. Perhaps this makes it easier to distinguish between neural responses from distinct parts of the brain such as the STG, which is associated with speech onset (Hamilton et al., 2018). In contrast, a larger $\rho$ for phase shift prediction could lead to representations that better discriminate neural synchrony information which is distributed across the brain rather than localised. As a result, a large proportion of the sensors in a MEG scanner should encode information about this feature.

### 4.3 Scaling Speech Decoding With Unlabelled Data

Here, we analyse generalisation as we increase the volume of unlabelled data, analysing scaling performance on downstream tasks. As before, we pre-train with the combined pretext tasks. Figure 3 shows ROC AUC as we increase the amount of unlabelled data in pre-training up to approximately 160 hours. For both tasks, pre-training with any amount of data is sufficient to beat chance and there is a clear improvement in accuracy as the amount of unlabelled data increases. For speech detection on Armeni et al. (2022), scaling appears logarithmic in log-space; for all others, ROC AUC improves log-linearly within the data regime we study. In any case, adding unlabelled data has improved generalisation. Notably, we have scaled far beyond the data regime of prior surgical and non-surgical work and yet performance has continued to scale. Thus, our self-supervision approach may remain useful as the volume of open data in the field continues to rapidly increase.

Our results also reveal several new and notable phenomena. Firstly, we scaled up the pre-training dataset by increasing the number of subjects. Since this led to consistent and almost monotonic improvements in downstream accuracy, our method is an exception to the common consensus that pooling subjects worsens generalisation. Secondly, as we pre-trained our model with a *different* dataset to those we fine-tuned on, our representation shows *cross-dataset generalisation*. This is particularly surprising as the Armeni et al. (2022), Gwilliams et al. (2023), and our pre-training dataset all use different scanners entirely. Performing well across these datasets indicates that, together, our architecture and pretext tasks successfully generate representations that are generalisable across heterogeneous scanners. Finally, we note that our pre-training dataset contained no language data whatsoever yet still improved downstream accuracy on language tasks. Remarkably, this shows that unlabelled brain data collected from *any* task (including those that are not linguistic) can be used to improve speech decoding performance.

Since the results show improvements on both downstream tasks, this indicates that our pretext tasks are sufficiently generic to produce representations that work with multiple speech decoding tasks while still generalising well on each task individually. This is generally a challenging trade-off to manage. However, we notice that in both tasks, the base accuracy is higher and the improvement in ROC AUC is steeper for Armeni et al. (2022). This is likely to be because this dataset has more within-subject data. The weaker results for Gwilliams et al. (2023) may be a consequence of the

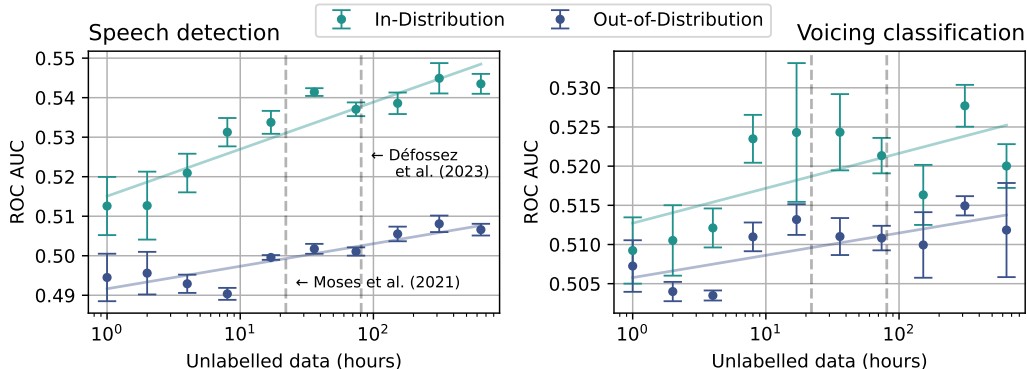

Figure 4: **Scaling unlabelled data improves novel subject generalisation.** We fine-tune on Gwilliams et al. (2023). When *in-distribution*, we evaluate on held-out sessions from subjects in the training set; when *out-of-distribution*, we evaluate on three held-out subjects. The solid lines are the best linear fits while the dashed lines show the amount of data used in prior surgical (Moses et al., 2021) and non-invasive (Défossez et al., 2023) work.

larger number of subjects with shorter intra-subject recordings and greater subject variation. These observations support the findings of other recent work such as Csaky et al. (2022).

### 4.4 SCALING UNLABELLED DATA IMPROVES GENERALISATION TO NOVEL SUBJECTS

In neuroimaging, brain data is generally highly variable across participants, leading to difficulty transferring models to novel subjects (Csaky et al., 2022). Whilst we have shown generalisation *across* subjects, here, we investigate whether we can generalise to *novel* subjects—an even more difficult challenge. This is critical in order to widely deploy speech BCIs for new patients. In this experiment, we fine-tune only on Gwilliams et al. (2023) and hold out three subjects with which we evaluate novel subject generalisation.

Figure 4 shows that scaling up the amount of unlabelled data used in pre-training not only improves accuracy on subjects previously seen, but also demonstrates a positive log-linear trend in performance for novel subjects. This indicates that scaling our method is an encouraging direction for resolving the challenges of subject variance faced by prior work. Moreover, as far as we are aware, this is the first result to demonstrate *novel* subject generalisation in speech decoding from MEG.

### 4.5 AGGREGATING UNLABELLED MEG DATASETS

To scale up unlabelled data further than individual studies, we must be able to combine many existing datasets. As a preliminary investigation, we combine two of the largest public MEG datasets: MOUS (Schoffelen et al., 2019) and Cam-CAN (Shafto et al., 2014; Taylor et al., 2017). In this section, we investigate how pre-training with these combined datasets affects downstream performance using the same experimental setup as Figure 3.

Table 2: **Combining unlabelled datasets shows signs of outperforming single studies.** We examine performance on the speech detection task. We see a small improvement when the datasets are combined on Gwilliams et al. (2023), but not Armeni et al. (2022).

| Pre-training dataset | Hours | Armeni | | | Gwilliams | | |
|---|---|---|---|---|---|---|---|
| | | ROC AUC | $t$ | $p$ | ROC AUC | $t$ | $p$ |
| Cam-CAN | 159 | **0.621** ± 0.003 | 36 | 4e−4 | 0.543 ± 0.003 | 13 | 3e−3 |
| MOUS | 160 | 0.605 ± 0.000 | 261 | 7e−6 | 0.543 ± 0.004 | 9 | 5e−3 |
| Cam-CAN + MOUS | 319 | 0.611 ± 0.003 | 40 | 3e−4 | **0.546** ± 0.002 | 20 | 1e−3 |

The results in Table 2 show, for the first time, that combining datasets can improve performance on downstream speech decoding tasks. It leads to better performance on Gwilliams et al. (2023) compared to pre-training on either dataset alone. Interestingly, this was not the case for Armeni et al. (2022) where pre-training on Cam-CAN alone performed best. Combined pre-training did, however, outperform training only on MOUS. It is surprising that pre-training on Cam-CAN was better than pre-training on MOUS when evaluating on Armeni et al. (2022) given that MOUS and Armeni et al. (2022) both used speech tasks and were acquired on the same MEG scanner. Cam-CAN, by contrast, did not use a speech task and was acquired on a different MEG scanner. We hypothesise that the better results for Cam-CAN are due to it being a cleaner dataset. During our experiments, we found that data quality, even among unlabelled data, can have a significant affect as artefacts in recordings disrupt learning.

While the combination of the two datasets includes far more hours of data than any prior work on deep learning with MEG, further work needs to be done to aggregate more datasets. Here, we were limited by compute budget. Increasing the number of datasets could enable the network to eventually always improve over the best singular dataset. Just as increasing the number of subjects (rather than only within-subject data) improves novel subject generalisation, a larger number of datasets may be key to scaling results when datasets are aggregated in pre-training.

### 4.6 LIMITATIONS

Although our results are significant in demonstrating a viable path forward to scale up speech BCIs, there remain a number of limitations to the present work. We focused here on two downstream tasks: speech detection and voice classification. Ultimately, we would like to expand this work to predict full transcripts from brain recordings (i.e. *brain-to-text*). This has been achieved with surgical data (Moses et al., 2021; Willett et al., 2023) but not yet convincingly with non-invasive methods like MEG or EEG (Jo et al., 2024). Speech detection has played an important role in the development of full brain-to-text in a surgical context (Moses et al., 2021) and we hope may play a similar role for non-invasive methods. Prior work has further used voice classification as a stand in for phoneme classification (Gwilliams et al., 2022), and we have been able to improve on these results here. In future work, we would like to expand this to all English phonemes. Secondly, while we have been able to demonstrate the utility of a few pretext tasks, we do not claim to have exhausted the full set of useful tasks. Rather, we conjecture that more useful pretext tasks remain to be found and believe a useful avenue of research will be into other input representations for brain recordings. For example, this paper did not make use of spatial features. Another limitation is our emphasis on heard speech over other types of speech, such as attempted or imagined speech. We hypothesise that the same methods presented here will generalise to these other varieties of speech, though this has yet to be shown. But, perhaps the biggest limitation of the present work is that, while it surpasses the amount of data used in other studies, it remains to be seen how much speech decoding tasks can be improved by scaling up the number of datasets used in training. In sharing this work now, we believe that the current proof of concept will be sufficiently impactful to the field as we continue to actively scale up the datasets that we can leverage.

## 5 CONCLUSION

Ultimately, solving speech decoding could transform the lives of patients with severe communication difficulties. This promise has not yet materialised because the field has been blocked by its inability to scale up data to leverage deep learning. Prior methods have been unable to aggregate data across different datasets, labels, or subjects to scale up because of heterogeneity in recording hardware, experiment design, and participants. A handful of studies have shown weak signals towards alleviating these issues. But until now, no one has developed a general solution. We provided a unified method that leverages unlabelled recordings data-efficiently using generic pretext tasks that shows that all of these problems can be solved. We verified this with experiments showing that our method not only scales with heterogeneous data but even generalises across datasets, subjects, and tasks. Our method unlocks the potential of the bitter lesson, providing a general method to exploit more computation by using more data. We implore the research community to employ the vast quantities of data and compute available to realise this potential. If scale is all you need in speech decoding, then the bitter lesson may not be so bitter.

## ETHICS STATEMENT

In this work, we use data from studies that involve human subjects (Armeni et al., 2022; Gwilliams et al., 2023; Shafto et al., 2014; Taylor et al., 2017; Schoffelen et al., 2019). These datasets are public, cited, and have their own ethical approvals. The documentation for these is available with the publications for the respective datasets.

While there are clear positive impacts, we acknowledge that insights from neural speech decoding research may not all be beneficial. Research in this field could enable paralysed patients to communicate freely and materially assist those with minor communication difficulty (e.g. stammering). As the technology matures, it could also enable new ways of communicating with others and interacting with devices without the risks of invasive surgical implants. Nevertheless, the maturity of this technology could also present potential negative societal impacts. For one, reading inner speech creates new concerns over data controls as this information is likely to be highly sensitive and personal to individuals. Given access to this technology, there is also the risk that bad actors could extract sensitive information from target individuals without consent. Moreover, there are possible long horizon effects associated with speech decoding research. Broad adoption of this technology could lead to the gradual erosion of privacy over inner speech within society. In addition, asymmetric effects, where some individuals or organisations can read inner speech but others are unable to, could worsen societal inequality. Within the scope of this paper, we mitigate risks associated with inner speech by focusing on decoding heard speech where there is low potential for abuse. Nonetheless, we acknowledge that this is still a stepping stone towards solving inner speech decoding.

## REPRODUCIBILITY STATEMENT

In the supplementary materials, we have provided an anonymised code repository with instructions for reproducing our main experiments. We also include details on experiment design and setup (Section 4.1 and Appendix A), hyperparameters (Appendix B), and compute (Appendix C). While we attempt to be exhaustive with these details, any information not found directly in the main body or appendices can be located in the supplementary materials.

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

## A EXPERIMENT DETAILS

We pre-train with non-overlapping sample windows from all subjects and sessions. We adjust the amount of unlabelled data used from Cam-CAN by increasing the number of subjects in the sequence 1, 2, 4, 8, 17, 36, 74, 152, 312, and 641, successively randomly selecting more subjects to include. Each seed uses a different set of subjects to reduce negative effects from outlier subjects.

When fine-tuning with Armeni et al. (2022), we hold out session 010 from all subjects during training and validation, using this for evaluation. Similarly, when fine-tuning with Gwilliams et al. (2023), we hold out session 1 from subjects 23, 24, 25, 26, and 27, using these sessions for evaluation only. As there is limited within-subject data in the latter dataset, we did not hold out a session from all subjects as before. For our novel subject experiments, we hold out subjects 1, 2, and 3 entirely and use the data for these subjects during evaluation. In Gwilliams et al. (2023), we note that they use four different tasks for each subject and their order is randomized between subjects. Both sessions for each task are repeats of the task. This means that while the recording itself is unseen, in this dataset, it is possible that heldout sessions use tasks that may have been seen in the training set.

In all experiments, we always fine-tune to completion (usually around 300 epochs), taking the test metric at the best validation loss (early stopping). We use three randomly selected seeds for each pre-training and corresponding fine-tuning run. For speech detection, since our encoder reduces the temporal dimension from 125 samples (the number of samples in a 0.5 second window with a sample rate of 250Hz) down to 5 embeddings, we downsample our speech detection labels to match using PyTorch's `torch.nn.functional.interpolate`. Therefore, each speech detection label represents a 0.1 second period of time.

# B  HYPERPARAMETERS

We conducted a search over hyperparameters of interest to optimise our self-supervised objectives and neural architecture. While these ablations indicated a theoretically ideal architectural configuration, in practice, we altered our final experimental architecture due to instabilities during training when data was scaled up. Our final architecture hyperparameters achieve a balance between the best values from our hyperparameter search and stable training. These values are detailed in Table 3.

Table 3: **Experimental hyperparameters.**

| Hyperparameter | Value |
|---|---|
| Window length (s) | 0.5 |
| $\rho$ (phase) | 0.5 |
| $\rho$ (amplitude) | 0.2 |
| $\{w_1, w_2, w_3\}$ | $\{1.0, 1.0, 1.0\}$ |
| $d_{\text{shared}}$ | 512 |
| $d_{\text{backbone}}$ | 512 |
| SEANet convolution channels | $(512, 512, 512, 512)$ |
| SEANet downsampling ratios | $(5, 5, 1)$ |
| FiLM conditioning dimension | 16 |
| Subject embedding dimension | 16 |
| Pre-training epochs | 200 |
| Optimizer | AdamW (Loshchilov & Hutter, 2019) |
| Learning rate | 0.000066 |
| Train ratio | 0.8 |
| Validation ratio | 0.1 |
| Test ratio | 0.1 |

## C COMPUTE RESOURCES

All experiments were run on individual NVIDIA V100 and A100 GPUs with up to 40GiB of GPU memory on a system with up to 1TiB of RAM. Each pre-training run with the maximum amount of pre-training data took approximately 200 hours (8.3 days). Fine-tuning following pre-training took up to another 12 hours. We estimate that we used approximately 3000 hours of compute for the final experimental runs, including hyperparameter searches. In total, over the course of developing this work from idea to final paper, we used around 10,000 hours of GPU compute.

## D LICENCES FOR DATASETS AND CODE

The Armeni et al. (2022) dataset is distributed under CC-BY-4.0 while the Gwilliams et al. (2023) dataset is distributed under the CC0 1.0 Universal licence. The Schoffelen et al. (2019) dataset is distributed with a RU-DI-HD-1.0 licence from the Donders institute. The licence for the Cam-CAN (Shafto et al., 2014; Taylor et al., 2017) dataset is unknown. The SEANet code adapted from Défossez et al. (2022) is distributed under the MIT licence, and the OSL library, which we use for preprocessing, is under the BSD-3-Clause licence.

