# OpenReview forum: "The Brain's Bitter Lesson: Scaling Speech Decoding With Self-Supervised Learning"
_ICLR.cc/2025/Conference — ICLR 2025 Conference Withdrawn Submission_

### Official Review · Reviewer_x8eN · 2024-11-04

**Soundness:** 3
**Presentation:** 3
**Contribution:** 2
**Rating:** 6
**Confidence:** 4

**Summary:**

- In short, the proposed approach uses self-supervised training on MEG data to improve downstream MEG-to-speech decoding. New pretraining tasks are proposed, with a focus on building a model that can learn from many different subjects and perform decoding on many different subjects. The authors find that the pretraining improves decoding over naive baselines and that increasing the amount of pretraining data leads to increased performance.

**Strengths:**

- There are other pretraining approaches for learning representations of brain data. But there is novelty in that the application (decoding speech from MEG) seems new.
- The approach is sound.
- The proposed pretraining tasks are all validated by ablation studies. The authors motivate the proposed tasks with neuroscience insights.
- The scaling results are promising for speech applications, especially for held-out subjects. I have concerns about novelty (see weaknesses) but these are tempered by the fact that at the very least, the proposed approach seems effective.

**Weaknesses:**

- While the application to MEG speech decoding is novel, there is limited novelty in the technique. Self supervised training for learning neural representations has been studied before, even for subject-generic representations, which is the main claim to novelty for this work. For example, [1,2,3,4] present subject generic approaches and use self-supervised training. And [5] presents a neuron-generic approach, but with the same principles. This is without mentioning all the work for foundational time-series models that exist in general. Given the existing work, it seems that the way for new work to distinguish itself is either to (1) show a downstream application of pretraining in a new domain or (2) show that the proposed pretraining tasks are different and more effective than others that exist. This work has (1) covered, but leaves (2) disputable. The proposed pretraining tasks share similarities with existing tasks (replace-discriminative learning from [4] or frequency phase forecasting from [1]). The formulations are not the exact same, but given that similar approaches exist, it seems that a new approach should justify why a different set of pretraining tasks is necessary.
- BIOT is used as a baseline. But it seems to be just the off-the-shelf version, without adaptation to the MEG domain (did I misunderstand? see questions section). So it's not fair to say that this approach is better than the complete BIOT self-supervised approach.
- The band prediction task seems unmotivated. The fact of presence/absence of information in a certain frequency band doesn't seem like semantic information. Importantly, it doesn't seem like the sort of task which encourages the model to learn anything specific about the distribution of neural activity. The ablation test shows that it is useful to performance, which is believable, since it could encourage some useful attention to low-level features, but there are many other plausible tasks that could fill this role.

## References
[1] Zhang, Daoze, et al. "Brant: Foundation model for intracranial neural signal." Advances in Neural Information Processing Systems 36 (2024).

[2] Yuan, Zhizhang, et al. "BrainWave: A Brain Signal Foundation Model for Clinical Applications" (2024)

[3] Joel Ye, et al. "Neural data transformer 2: multi-context pretraining for neural spiking activity. Advances in Neural Information Processing Systems (2024).

[4] Donghong Cai, et al. "Mbrain: A multi-channel self-supervised learning framework for brain signals. In Proceedings of the ACM SIGKDD Conference on Knowledge Discovery and Data Mining (2023).

[5] Le, Trung, and Eli Shlizerman. "Stndt: Modeling neural population activity with spatiotemporal transformers." Advances in Neural Information Processing Systems 35 (2022).

**Questions:**

- What is the nature of the subject conditioning on line 207? Is it a concatenated prompt vector? Some sort of fine-tuning?
- What is the downsampling rate from $t$ to $\tau$ (line 203)?
- Eq. 4 describes weightings $w_i$. Is anything other than a uniform weighting used?
- Line 326 suggests that BIOT was taken off-the-shelf (?) But Line 363 suggests that it was pre-trained on Cam-CAN?

---

> ### Author Response · Authors · 2024-11-19
> **Response to Reviewer x8eN**
>
> Thank you for taking the time to review our work.
>
> We first highlight that we have achieved substantially stronger speech detection results since submission.
>
> | Method | AUROC | F1 (macro) |
> | :---- | :---- | :---- |
> | **Ours (new)** | **0.700** \+/- 0.002 | **0.790** \+/- 0.003 |
> | Ours (submission) | 0.621 \+/- 0.003 | 0.670 \+/- 0.011 |
>
> To achieve this, we used 1s input windows (instead of 0.5s) to resolve lower frequencies in the signal and applied a 1D gaussian filter (with standard deviation 9) to our predictions (reducing false positives).
>
> Remarkably, this new result even matches the AUROC quoted in [A, Table 2] who use *intracranial* data from heard speech. We achieved this score with non-invasive data (which is typically substantially more difficult to decode due to low signal-to-noise ratio). This result is only on Armeni due to time/compute limitations in the discussion period but we will re-run everything in full for the camera-ready version.
>
> Please find below our responses to your specific questions and concerns:
>
> > Self supervised training for learning neural representations has been studied before, even for subject-generic representations
>
> Thank you for bringing [1, 2, 3, 4, 5] to our attention. We have cited these in the revised draft. What makes our work different to these is that we optimise our self-supervision method specifically for *speech decoding* tasks and using domain specific transformations (cf. domain agnostic tasks like next token prediction). This informed how we designed each one of our pretext tasks, using knowledge from the neuroscience literature on speech. We note that our work does not only generalise across subjects but also across data collected from different studies, and we show that it is possible to generalise from one pre-training dataset to multiple others, even though they were collected with different subjects and hardware profiles.
>
> > show that the proposed pretraining tasks are different and more effective than others that exist
>
> We agree that having more comparisons will be helpful to demonstrate point (2). Thank you for noting this. We have added BrainBERT [A] as an additional comparison baseline. We note that their pre-training approach was designed, like ours, to extract speech features as they evaluated with heard speech.
>
> Here are the results:
>
> | Method | AUROC | F1 (macro) |
> | :---- | :---- | :---- |
> | **Ours** | **0.705** \+/- 0.003 | **0.801** \+/- 0.011 |
> | BrainBERT \[A\] | 0.556 \+/- 0.007 | 0.761 \+/- 0.005 |
>
> For both methods, we pre-trained on a subset of CamCAN (36 subjects) and fine-tuned on Armeni subject 1. For time/compute reasons, we have not been able to run this on the other subjects yet. But we will have run this in full for the camera-ready version.
>
> > BIOT is used as a baseline. But it seems to be just the off-the-shelf version, without adaptation to the MEG domain
>
> We actually pre-train BIOT with MEG (CamCAN) in full as we did with our own method; it is *not* the off-the-shelf version. Thank you for highlighting that this was unclear. We have now changed the wording in the manuscript on line 326 to “pre-trained on CamCAN” to make this clear.
>
> > The band prediction task seems unmotivated. The fact of presence/absence of information in a certain frequency band doesn't seem like semantic information.
>
> With respect, we disagree with this assertion. By filtering and predicting missing speech-relevant frequency bands, we instil a sensitivity for those frequencies into the model’s learned representations. It is not the presence/absence of frequency bands that is helpful, but sensitivity in the representation that brings about better separability in the representation space. This is useful for speech decoding because, for example, there is higher power in low-frequency neural responses than high-frequency during speech onset than not [B, Fig. 2.2C, Fig 3.6, Fig B.7b]. Of course, we also show empirically that this pretext task helps with the downstream tasks (Table 1).
>
> > What is the nature of the subject conditioning on line 207? Is it a concatenated prompt vector? Some sort of fine-tuning?
>
> It is a learned vector concatenated to the input or latents.
>
> > What is the downsampling rate from to (line 203)?
>
> The downsampling ratio is 0.04.
>
> > Is anything other than a uniform weighting used?
>
> Sadly no. Pre-training is very computationally expensive, so we did not have a chance to explore this here, but hope to tune these weights in future work.
>
> Thank you again for taking the time to review our work. Please let us know if there are any other points we can address to raise your score.
>
> [A] Wang, C., Subramaniam, V., Yaari, A., Kreiman, G., Katz, B., Cases, I. and Barbu, A., 2023, August. BrainBERT: Self-supervised representation learning for intracranial electrodes. In International Conference on Learning Representations. ICLR.
>
> [B] Mantegna, F., 2024. The Temporal Dynamics of Speech Motor Control (Doctoral dissertation, New York University).

---

> ### Author Response · Authors · 2024-11-24
> **Follow-up**
>
> We are following up to check if we have addressed your main concerns. We would be happy to provide clarifications for any additional questions that you may have.

---

> ### Comment · Reviewer_x8eN · 2024-11-25
>
> I appreciate the author's time in answering my questions! I am still leaning towards the accept side, but some things are still unclear:
> - > By filtering and predicting missing speech-relevant frequency bands, we instil a sensitivity for those frequencies into the model’s learned representations
>   - Is the task to predict the *signal* that is omitted from a band $\omega$, or to predict the $\omega$ label itself, i.e., $\gamma$, $\theta$, etc. ?
> - > We actually pre-train BIOT with MEG (CamCAN) in full as we did with our own method
>   - Thank you for clarifying this! This does help increase my confidence.
> - > To achieve this, we used 1s input windows (instead of 0.5s) to resolve lower frequencies in the signal and applied a 1D gaussian filter
>   - One important thing to check: can we rule out that this is not due to data leakage? That is, are the train and test samples all being sampled from one time contiguous block of data? In which case, extending the window could cause train and test samples to overlap in time with each other.
> - > What makes our work different to these is that we optimise our self-supervision method specifically for speech decoding
>   - The motivations for the self-supervised loss given in the text were nice, but the losses themselves seem very generic. Is there a BCI task for which phase, frequency, and amplitude sensitivity would not be relevant? In fact, these are problems which general time series foundation models seek to solve (e.g. data augmentations for [Chronos](https://arxiv.org/pdf/2403.07815) and [Moirai](https://arxiv.org/pdf/2402.02592)).

---

> ### Author Response · Authors · 2024-11-25
> **Response to further questions**
>
> Thank you for your response and further questions!
>
> > Is the task to predict the signal that is omitted from a band [...] or to predict the label itself
>
> The task is to predict the label itself. We have revised the wording in the paper to make this clearer. Thank you. Discriminating between the frequency bands instills sensitivity to these bands and frequencies. Early in this work, we also considered the generative task of predicting the missing signal itself. However, we concluded that this is unlikely to be effective at the scale of unlabelled MEG data. It would be similar to a masking objective like the one used in [A], which we have found does not lead to downstream performance as strong as our approach does with MEG.
>
> > One important thing to check: can we rule out that this is not due to data leakage? [...]
>
> We are aware of the prevalence of data leakage---especially in neuroscience work [C]. We were careful to avoid such leakage and nonsense correlations by using separate recording sessions (taken at different times/days) for the test sets. This ensures the strongest siloing between our train and test data. This is a latent issue in the field and we have thought considerably about it in our experiment design. For example, in Armeni, we used sessions 1-9 as train/val and session 10 separately for test. There is no overlap in time. Thank you for raising this and we will highlight the care we have taken to avoid leakage in the paper.
>
> > Is there a BCI task for which phase, frequency, and amplitude sensitivity would not be relevant?
>
> We agree that these are generally useful properties for a model to be sensitive to for BCI tasks. While motivated by speech decoding, our tasks can potentially apply more generically. This is an advantage of our approach, potentially making it applicable beyond speech decoding and beyond MEG (e.g. to EEG or multi-modal brain signals). We are planning to explore this in future work. For completeness, we note that the symmetric question ("do tasks useful for other BCIs help speech BCIs?") does not hold. For example, time-reversal data augmentations that are useful in sleep studies [D] would corrupt our time-dependent speech signal.
>
> > [...] In fact, these are problems which general time series foundation models seek to solve
>
> While our tasks could be relevant, we do not think they would necessarily always be effective in general. It is not clear which frequencies are relevant for a generic task, for example. We also do not know whether phase or scale relationships between variates generally encode useful information (in the case of Chronos, they focus only on univariate time series). We think the value of these tasks will depend on the domain they are applied to---but this could be beyond brain data as you suggest.
>
> We hope this answers your remaining questions. Please let us know if you have any more concerns. Thank you again for your interest in our work!
>
> [C] Harris, K.D. (2020). Nonsense correlations in neuroscience. bioRxiv.
>
> [D] Rommel, C., Paillard, J., Moreau, T., & Gramfort, A. (2022). Data augmentation for learning predictive models on EEG: a systematic comparison. Journal of Neural Engineering, 19.

---

> ### Author Response · Authors · 2024-11-27
> **Response to further questions (part 2)**
>
> Thank you for your patience. In addition to the above comment, we wanted to highlight the following:
>
> Your suggestions for additional baselines (such as frequency-phase forecasting [1]) helped us crystallise another essential property of our tasks that makes them unique from generic time series unsupervised learning (as well as much of the brain decoding literature). In attempting to implement these tasks, we were reminded that our tasks are crucially **sensor-agnostic**, meaning that they are designed to not require predicting properties specific to individual sensors. This (1) allows pooling heterogeneous datasets in pre-training, facilitating scaling up, and (2) harmonises our latent representation and enables our method to generalise well to different hardware with different numbers of sensors. We showed this by pre-training on one dataset and generalising to two different datasets. Tasks such as the ones you have suggested can not be used in our method because they are **not sensor-agnostic**. Taking frequency-phase forecasting as an example, it requires predicting the dominant frequency and instantaneous phase (which are properties of each individual sensor). Doing so would not allow multiple datasets with different numbers of sensors to be pooled in pre-training. Thank you for your suggestions. They have not only helped make the contributions clearer but also assisted us in gaining clarity ourselves.
>
> We hope this response and the one above have helped further clarify things. Do you have any additional concerns we may help alleviate to raise your score?

---

> > ### Comment · Reviewer_x8eN · 2024-12-01
> >
> > I thank the author for their response. I will keep my score and continue to recommend accept.

---

### Official Review · Reviewer_8GQ2 · 2024-11-04

**Soundness:** 3
**Presentation:** 3
**Contribution:** 2
**Rating:** 5
**Confidence:** 5

**Summary:**

The manuscript reports on an attempt to improve MEG decoding by pretraining on unlabelled data.  In particular, the authors focus on two binary classification tasks: speech detection and voicing detection.  In the pretraining (pretext) task, the input MEG signal is either phase shifted on randomly selected sensors (by a single, random, discrete phase); amplitude scaled at random sensors (by a single, random, discrete scalar); or bandstop filtered at a single random frequency band.  The network is trained to classify which phase shift, amplitude scaling, or filter was applied.  The authors report that pretraining improves performance on the binary-classification tasks, with simultaneous training on all three pretexts yielding the best results.  Performance on data from subjects not in the fine-tuning training set also improves with number of data used in pretraining.

**Strengths:**

The result is statistically significant and novel in the area of MEG, at least to this reviewer's knowledge.  The pretext tasks are potentially applicable to other neural recording modalities (e.g., scalp and intracranial EEG).

**Weaknesses:**

(1) The amount of information being extracted from the MEG signals under even the best models is very low.  The best AUC achieved on these binary-classification tasks is ~0.62 (chance being 0.5).  This kind of performance on speech *detection* does not inspire confidence in the ability of MEG to scale up to actual decoding of words (to say nothing of the fact that it is perceived, rather than attempted, speech that is being decoded).

The authors describe performance as scaling linearly in the log of the number of hours of data, which might seem to give some hope for BCI application.  But (eyeballing Fig. 4) the slope is around 0.025 in AUC per order of magnitude, at which rate something like 10^32 hours of data would be required to reach 90% AUC on *speech detection* (assuming the linearity held out that long).

And these are the stongest results in the MS.  The use of multiple datasets in pretraining does not improvement classification performance on the Armeni data at all (Table 2), and the improvement from the Gwilliams data does not appear likely to be statistically significant (the authors should certainly test this).

In short, the results are not encouraging for the use of MEG in speech decoding.

(2) There are lots of other models attempting to use self-supervised learning on EEG and the like (i.e., in addition to BIOT), e.g.,

Brant (https://proceedings.neurips.cc/paper_files/paper/2023/file/535915d26859036410b0533804cee788-Paper-Conference.pdf)
BrainBERT (the authors reference this study)
MMM (https://openreview.net/pdf?id=hiOUySN0ub)
BENDR (https://www.frontiersin.org/journals/human-neuroscience/articles/10.3389/fnhum.2021.653659/full)
LaBraM (the authors reference this study)
EEGFormer (https://arxiv.org/abs/2401.10278)

I don't think the authors need to have compared to all of these, but since BIOT seems to have failed entirely (chance performance), surely they could have tried one of these others instead.


MINOR
There are at least two ECoG studies of speech decoding that do train single models on multiple subjects' data: Anumanchipalli et al., Nature, 2019; and Makin et al., Nature Nueroscience, 2020.

**Questions:**

I take it Table 1 is for speech detection (as opposed to voicing).  (I don't see this is the caption or text but I may have missed it.)  If that's right, how does voicing detection compare?  Likewise for Table 2 (although this is clearly labeled "speech detection").

Are the differences between "band-only" pretraining and "all tasks" statistically significant?  (This seems unlikely, at least for the Gwilliams data, since N=3.)

---

> ### Author Response · Authors · 2024-11-19
> **Response to Reviewer 8GQ2**
>
> Thank you for your time and effort. Please find below our responses to your specific concerns:
>
> > The amount of information being extracted from the MEG signals under even the best models is very low
>
> We have now achieved substantially stronger speech detection results since submission.
>
> | Method | AUROC | F1 (macro) |
> | :---- | :---- | :---- |
> | **Ours (new)** | **0.700** \+/- 0.002 | **0.790** \+/- 0.003 |
> | Ours (submission) | 0.621 \+/- 0.003 | 0.670 \+/- 0.011 |
>
> To achieve this, we used 1s long input windows (instead of 0.5s) to be able to resolve lower frequencies in the signal and applied a 1D gaussian filter (with standard deviation 9) to our predictions (reducing false positives).
>
> Remarkably, this new result even matches the AUROC quoted in [A, Table 2] who use *intracranial* data from heard speech. We achieved this score with non-invasive data (which is typically substantially more difficult to decode due to low signal-to-noise ratio). This result is only on Armeni due to time/compute limitations in the discussion period but we will re-run everything in full for the camera-ready version.
>
> > (to say nothing of the fact that it is perceived, rather than attempted, speech that is being decoded)
>
> We wonder if the reviewer is confusing attempted with imagined speech. Attempted speech is much closer to overt speech and can only be produced by patients who are at least partially paralysed, as it is overt speech in the general population. In recent invasive works, patients have had residual movement and were mouthing or vocalising during attempted speech. We agree that imagined speech is significantly harder to decode then heard speech, but it is unclear that this is true for attempted speech. The similarity of attempted speech to overt speech (e.g. in motor cortex) suggests that attempted speech may be easier to decode.
>
> > The use of multiple datasets in pretraining does not improvement classification performance
>
> Unfortunately we did not acquire enough datasets to pool in this case to overcome domain diversity in all cases, so our main contribution in this section is a method that supports pooling. We included this experiment as it tells follow up work that more than two datasets are likely to be required in pre-training if pooling. We discuss this as a limitation and encourage follow up works to scale up datasets. Pooling heterogeneous datasets in this domain is notoriously challenging. We see our pre-training workflow as a promising step, but further work will be needed before we understand the subtleties of mixing brain data.
>
> > surely they could have tried one of these others instead.
>
> Thank you for bringing these to our attention. We have cited these in the revised draft and added BrainBERT [A] as an additional baseline. We note that their pre-training approach was designed, like ours, to extract speech features as they evaluated with heard speech (rather than being a more generic model).
>
> | Method | AUROC | F1 (macro) |
> | :---- | :---- | :---- |
> | **Ours** | **0.705** \+/- 0.003 | **0.801** \+/- 0.011 |
> | BrainBERT \[A\] | 0.556 \+/- 0.007 | 0.761 \+/- 0.005 |
>
> For both, we pre-trained on a subset of CamCAN (36 subjects) and fine-tuned on Armeni subject 1. We will have results for the other subjects in the camera-ready version.
>
> > There are at least two ECoG studies of speech decoding that do train single models on multiple subjects' data: Anumanchipalli et al., Nature, 2019; and Makin et al., Nature Nueroscience, 2020.
>
> We have now cited these. To clarify, this is similar to [B], but with supervised learning in invasive data rather than unsupervised learning in non-invasive data.
>
> > I take it Table 1 is for speech detection (as opposed to voicing). (I don't see this is the caption or text but I may have missed it.) If that's right, how does voicing detection compare?
>
> We have amended the table caption to make it clear that this is speech detection---thank you. We did not run voicing as we focused on speech detection as our primary task. The idea was to add voicing as an additional, more challenging task afterwards. But would you like us to run this experiment?
>
> > Are the differences between "band-only" pretraining and "all tasks" statistically significant? (This seems unlikely, at least for the Gwilliams data, since N=3.)
>
> It is not statistically significant with the current N=3. We are running this now with more seeds.
>
> Thank you again for taking the time to review our work. Please let us know if there are any other points we can address to raise your score.
>
> [A] Wang, C., Subramaniam, V., Yaari, A., Kreiman, G., Katz, B., Cases, I. and Barbu, A., 2023, August. BrainBERT: Self-supervised representation learning for intracranial electrodes. In International Conference on Learning Representations. ICLR.
>
> [B] Défossez, A., Caucheteux, C., Rapin, J., Kabeli, O. and King, J.R., 2023. Decoding speech perception from non-invasive brain recordings. Nature Machine Intelligence, 5(10), pp.1097-1107.

---

> ### Author Response · Authors · 2024-11-24
> **Follow-up**
>
> We would like to follow up to check if we have addressed your main concerns. We would be happy to provide clarifications for any additional questions.

---

> > ### Comment · Reviewer_8GQ2 · 2024-11-25
> >
> > > We have now achieved substantially stronger speech detection results since submission.
> >
> > Very good--these are more convincing.  Can you give a new (improved) estimate of how many hours of unlabelled data will be required to reach (say) 90% AUC for speech detection?
> >
> > > We wonder if the reviewer is confusing attempted with imagined speech
> >
> > My point is that the MS decodes *perceived* rather than *produced* (or attempted) speech!  How could someone who can't speak communicate if we decode only the speech that he hears?  Is the authors' view that the results will carry over to produced/attempted speech (even though the decoding will be based on motor, rather than temporal, cortex)?  To my knowledge, there is not much work on decoding speech production with MEG, presumably because it is very sensitive to movement artifacts.  Or do the authors have in mind some other path from perceived speech to a BCI?
> >
> > > Unfortunately we did not acquire enough datasets to pool in this case to overcome domain diversity in all cases
> >
> > I share the authors' underlying intuition that, with enough data sets, pooling will eventually become helpful.  But this MS has not shown this, which is particular problematic since it seems (Figs. 3, 4) that an enormous number of hours of unlabelled data are required to achieve passable speech detection---too many hours to be collected from one experiment alone.  (And is the putative improvement on even the Gwilliams dataset statistically significant?)
> >
> > > added BrainBERT
> >
> > Thanks; this is a useful comparison.  Do the authors have results for any other subjects at this point?  Also, the BrainBERT F1 score appears close to the authors' (new) results, but its AUROC is worse.  Can the authors provide some insight here?  (Is this due to imbalance classes?)  Can you say something about why you think your method outperforms BrainBERT?
> >
> > > But would you like us to run this experiment?
> >
> > Yes, it would be useful to see results for voicing detection.
> >
> > > It is not statistically significant with the current N=3. We are running this now with more seeds.
> >
> > Please do report if you find significant differences here.  If not, the authors will need to restate their claims.  (It would still be an interesting finding if amplitude and phase prediction are helpful by themselves but do not provide any additional improvement over band prediction.)
> >
> > In light of the improved detection results, I will raise my score one point, but I still can't see a path from these results to speech decoding from MEG:  The detection is not great; the results are for perception rather than production; pooling over experiments doesn't seem to work; voicing classification is barely about chance (but is it better with the new windowing?); the amplitude and phase predictions don't add anything significant beyond the band prediction; and so far we only know that the method is better than an existing method (BrainBERT) on one test subject.

---

> ### Author Response · Authors · 2024-11-26
> **Response to further questions**
>
> Thank you for your further questions.
>
> > Can you give a new (improved) estimate of how many hours of unlabelled data will be required to reach (say) 90% AUC
>
> Yes. To make an estimate, we collected two additional data points using our new method to make three data points. The first with no pre-training, the second using 9 hours of pre-training, and the third using 160 hours of pre-training. Fitting a power law curve to these points yielded y \= 0.12x^0.14 \+ 0.4861. Extrapolating this curve to 0.8 AUC suggests 1000 hours of unlabelled data will be required; to reach 0.9 AUC, approximately 7000 hours of unlabelled data would be required. Note that this estimate may have very high variance given the few datapoints. We will collect more data (updating Figure 3\) to improve it.
>
> > How could someone who can't speak communicate if we decode only the speech that he hears?
>
> For practical reasons, such as a lack of sufficient inner speech data, we have turned to heard speech to develop methods that might be generalised to inner speech later. There is lots of evidence that imagined speech shares neural representations with speech comprehension. Imagined speech can be thought of as "auditory imagery". For reference, see \[C\]. Decoding heard speech is also interesting in itself, as evidenced by recent and influential work on it (e.g. \[B\]). We apologise and will clarify if the paper sounds like a working BCI is the only motivation for decoding heard speech.
>
> >  is the putative improvement on even the Gwilliams dataset statistically significant?
>
> Yes, the improvement is indeed statistically significant for both speech detection (p \= 0.004) and voicing (p \= 0.018). We note that the Gwilliams dataset is particularly challenging because it has very little within-subject data for its 27 participants.
>
> > Do the authors have results for any other subjects at this point?
>
> Yes, here are results for subjects 2 and 3:
>
> | Method | AUROC | F1 (macro) |
> | :---- | :---- | :---- |
> | Ours (Sub. 2\) | 0.720 | 0.795 |
> | BrainBERT (Sub. 2\) | 0.638 | 0.77 |
> | ---- | ---- | ---- |
> | Ours (Sub. 3\) | 0.724 | 0.802 |
> | BrainBERT (Sub. 3\) | 0.609 | 0.759 |
>
> We will continue to collect more results (i.e. Gwilliams) and scale up the pre-training data.
>
> > \[...\] the BrainBERT F1 score appears close to the authors' (new) results, but its AUROC is worse. \[...\] (Is this due to imbalance classes?)
>
> Yes, there is class imbalance. Approximately 61% of the samples are positive (speech) vs 39% negative (not speech). This can vary somewhat by subject also.
>
> > Can you say something about why you think your method outperforms BrainBERT?
>
> We think the reasons are twofold: firstly, BrainBERT focuses on a relatively generic pre-training task (masked spectrogram in-filling). While the focus of their evaluation is on speech decoding, their pre-training task may not be sufficiently motivated towards learning features that are sensitive to those properties relevant to neural responses to speech. Second, our intuition is that this infilling task is more difficult than our classification tasks, thus requiring substantially more data to learn. Another way to view this is that our tasks are more data-efficient at the current data scale. Whereas they design their method to train with comparatively less noisy intracranial data, MEG is a more challenging modality due to lower signal-to-noise. The unlabelled MEG data we use appears to not be enough to learn as effectively with their pre-training task.
>
> > I still can't see a path from these results to speech decoding from MEG
>
> There doesn't need to be a path to a BCI for the paper to be valuable. There has been lots of interesting work on decoding heard speech in itself (e.g. \[B\]). If we have given the impression that this is the only motivation or value of the work, then we apologise.
>
> For BCI applications, we are primarily interested in inner speech. There are two ways in which the current work advances an inner speech BCI. First, it develops general methods that could be used in that domain. We can't show it here in this paper without inner speech data, but we hypothesise that the same kind of approach (pretext training) will generalise from heard speech to inner speech. So developing the methods is valuable.
>
> A second way in which the present paper might help to advance an inner speech BCI is through the similarity of inner speech and heard speech. There is a substantial literature on the similarities between the two kinds of speech (e.g. \[C\]).
>
> Thank you again for your interest in this work. We are working on addressing all your points, so please let us know if you have further concerns.
>
> \[C\] Peter Langland-Hassan (2018) "From Introspection to Essence: The Auditory Nature of Inner Speech" in Inner Speech: New Voices, Oxford University Press.

---

> > ### Comment · Reviewer_8GQ2 · 2024-11-27
> >
> > Thanks for your in-depth reply.  But I am still quite confident in my rating (officially 5 because 4 wasn't an option), for the reasons given above.

---

### Official Review · Reviewer_zv6c · 2024-11-04

**Soundness:** 3
**Presentation:** 3
**Contribution:** 3
**Rating:** 6
**Confidence:** 3

**Summary:**

The manuscript describes a pipeline for self-supervised learning for  Magnetoencephalography (MEG) datasets. They introduce three pretext tasks predicting hand-designed perturbations of the input signal, predicting either 1) which frequency band was removed via bandstop filter (band prediction); 2) how much the phase was shifted in a random subset of sensors (phase shift prediction);  3) how much the amplitude was scaled in a random subset of sensors (amplitude scale prediction).
They use the medical Cam-CAN MEG dataset as the unlabelled self-supervised dataset and evaluate on speech detection and voice classification as downstream tasks for which the self-supervised-trained model is finetuned. Results show improvement over a linear baseline, no pretraining and a reimplemented self-supervised baseline from prior work. THey find combining all three pretext tasks outperforms any single one. Furthermore, increasing unlabelled pretraining data improves downstream performance.

**Strengths:**

* Presentation is good, writing is understandable, if a bit longwinded at times, figures are professional and everything is legible
* motivation and main idea is straightforward and clear
* results for varying unlabelled dataset sizes are interesting

**Weaknesses:**

The scientific value of such a manuscript is highly dependent on the presented comparisons to other works as well as the comparability of the current results for future works.

For the failed BIOT experiments, is it possible to obtain a pretrained BIOT model and apply it, potentially to a subset of electrodes?

For the   [[2405.18765] Large Brain Model for Learning Generic Representations with Tremendous EEG Data in BCI](https://arxiv.org/abs/2405.18765)  work, a comparison to this may be very helpful. I think (not sure) they also have pretrained models available.

Is it possible to compare to the to existing results for the downstream tasks like the Defoussez works for example?

In general, reporting further metrics other than AUROC that allow more direct comparisons to other speech decoding works would be crucial to better understand the performance of the proposed approach.

Minor: [[1911.05419] Self-supervised representation learning from electroencephalography signals](https://arxiv.org/abs/1911.05419)  might be  another relevant work utilizing different pretext tasks.

**Questions:**

The phase perturbation, how is it done? Is it applied in the frequency domain to all frequency bins? Did you consider only shifting randomly selected bands like gamma alpha etc.? Same for the amplitude scaling.

The subject conditioning, how does it work exactly, it is only trained during pretraining as far as I understand from the figure? So when you have a new dataset fora  new subject what exactly do you need to compute?

---

> ### Author Response · Authors · 2024-11-19
> **Response to Reviewer zv6c**
>
> Thank you for your time and effort in reviewing our work.
>
> We have achieved substantially stronger speech detection results since submission.
>
> | Method | AUROC | F1 (macro) |
> | :---- | :---- | :---- |
> | **Ours (new)** | **0.700** \+/- 0.002 | **0.790** \+/- 0.003 |
> | Ours (submission) | 0.621 \+/- 0.003 | 0.670 \+/- 0.011 |
>
> To achieve this, we used 1s input windows to resolve lower frequencies in the signal and applied a 1D gaussian filter (with standard deviation 9) to our predictions (reducing false positives).
>
> Remarkably, this new result matches the AUROC quoted in [A, Table 2] who use *intracranial* data from heard speech. We used non-invasive data (which is typically substantially more difficult to decode due to low signal-to-noise ratio). This result is only on Armeni due to compute limitations in the discussion period but we will re-run in full for the camera-ready version.
>
> > The scientific value of such a manuscript is highly dependent on the presented comparisons
>
> We agree. We have added BrainBERT [A] as an additional comparison. We note that their pre-training approach was designed, like ours, to extract speech features as they evaluated with heard speech (rather than being a more generic model like other approaches).
>
> Here are the results:
>
> | Method | AUROC | F1 (macro) |
> | :---- | :---- | :---- |
> | **Ours** | **0.705** \+/- 0.003 | **0.801** \+/- 0.011 |
> | BrainBERT \[A\] | 0.556 \+/- 0.007 | 0.761 \+/- 0.005 |
>
> For both methods, we pre-trained on a subset of CamCAN (36 subjects) and fine-tuned on Armeni subject 1. We will have results for the others in the camera-ready version.
>
> > is it possible to obtain a pretrained BIOT model and apply it, potentially to a subset of electrodes?
>
> Thanks for the suggestion. Here are the speech detection results on Armeni using the first 18 sensors.
>
> | Method | AUROC | F1 (macro) |
> | :---- | :---- | :---- |
> | **Ours** | **0.700** \+/- 0.002 | **0.790** \+/- 0.003 |
> | Pre-trained BIOT (6 EEG datasets) | 0.615 \+/- 0.002 | 0.661 \+/- 0.003 |
>
> We also agree that LaBraM would make a good comparison but no model weights are available and we were unable to train it successfully.
>
> > Is it possible to compare to the to existing results for the downstream tasks like the Defoussez works for example?
>
> The method in [B] requires paired brain and audio data, which is relatively rare so does not scale as easily as ours which only requires brain data. The downstream task in [B] matches arbitrary 3 second segments of audio and brain. This is much longer than most words and could be picking up on lots of other things (e.g. patterns of silence). In any case, [B] only reports top 10 accuracies for the audio/brain matching task and not for word classification. Because it requires paired audio and brain data, it is unlikely to be a useful task for future BCIs.
>
> > In general, reporting further metrics other than AUROC that allow more direct comparisons
>
> Thank you for making this point. We have provided F1 (macro) scores in our new results (above). We can also provide these for all other results in the camera-ready version. We apologise that we could not already provide those now, due to time/compute constraints. Are there any other metrics you think will best allow for comparison to specific papers? We followed [A] in choosing AUROC in the initial submission.
>
> > Minor: [1911.05419] Self-supervised representation learning from electroencephalography signals might be another relevant work utilizing different pretext tasks.
>
> Thanks for bringing this to our attention. We agree, temporal contrastive tasks could also be relevant for developing SSL methods for speech. We have added this citation in the revised draft.
>
> > The phase perturbation, how is it done? Is it applied in the frequency domain to all frequency bins? Did you consider only shifting randomly selected bands like gamma alpha etc.? Same for the amplitude scaling.
>
> Yes, the phase shift is applied in the frequency domain. We did not try shifting only selected bands, but it would be interesting to explore this. Thank you.
>
> > The subject conditioning, how does it work exactly
>
> For novel datasets/subjects, we use the average of trained subject embeddings. Interestingly, we found that the subject conditioning only helped in cross-subject generalisation rather than novel subject generalisation. We have been thinking about how to improve this in future work.
>
> Thank you again for taking the time to review our work. Are there any other points we can address that might raise your score?
>
> [A] Wang, C., Subramaniam, V., Yaari, A., Kreiman, G., Katz, B., Cases, I. and Barbu, A., 2023, August. BrainBERT: Self-supervised representation learning for intracranial electrodes. In International Conference on Learning Representations. ICLR.
>
> [B] Défossez, A., Caucheteux, C., Rapin, J., Kabeli, O. and King, J.R., 2023. Decoding speech perception from non-invasive brain recordings. Nature Machine Intelligence, 5(10), pp.1097-1107.

---

> ### Author Response · Authors · 2024-11-24
> **Follow-up**
>
> We wanted to follow up to check if we have addressed your concerns. If you have further questions or details, we would be happy to provide additional clarifications.

---

> ### Comment · Reviewer_zv6c · 2024-11-25
> **Thanks, Comparable Downstream Tasks would be valuable in the future**
>
> Thank you very much for the answers and additional experiments that definitely improve the value of this paper in my view. I have increased my score. There is not really time for this now, but in general I think including downstream tasks for which comparable results exist in the literature such as the tasks from [B] would make the manuscript more valuable. Even if those comparisons show that this approach underperforms another approach on those tasks, at least one would have a clearer picture.
> Is my understanding correct that these downstream tasks (speech detection, voicing classification), nobody has done in this way and that is why there is no comparison?

---

> > ### Author Response · Authors · 2024-11-25
> > **Thank you for your suggestions**
> >
> > We're glad to hear that our answers and additional experiments improved the value of the paper. Thank you for raising your score. Yes, no other paper we are aware of has benchmarked perceived speech MEG on speech detection or voicing classification. This is the reason that we had to train other models ourselves for comparison. Thanks for highlighting that having more comparisons would give a clearer picture in this case. We will focus on adding further comparisons for the final revision (e.g. the task from [B]). Thanks once again for your interest in our work!

---

### Author Response · Authors · 2024-12-03
**Thanks to Reviewers + Additional Dataset Aggregation Results**

We would like to thank all the reviewers for taking the time and effort to review our work and respond to our comments. Your feedback was valuable and made us revise the paper in ways that significantly improved the thoroughness of our work.

While it is too late for reviewers to respond further, we want to continue to demonstrate our commitment to taking into account their feedback. The remaining experimental concern we left unanswered in our replies was to demonstrate improvement in downstream generalisation through aggregating multiple unlabelled datasets. We have now achieved this, too.

| Pre-training Datasets | AUROC |
| :---- | :---- |
| Gwilliams | 0.559 \+/- 0.004 |
| MOUS | 0.530 \+/- 0.010 |
| Cam-CAN | 0.553 \+/- 0.0001 |
| Gwilliams \+ MOUS \+ Cam-CAN | **0.601** \+/- 0.001 |

In the above experiment, we combined 10-hour subsets of Gwilliams, MOUS, and Cam-CAN for unlabelled pre-training and evaluated it on speech detection with Armeni subject 1. Here, we have shown that we can outperform individual datasets by aggregating unlabelled data. We made only minor changes to achieve this: 1) use deeper dataset adapters rather than dataset-conditional linear layers and 2) combine more than two datasets. As we hypothesised, having more datasets has helped make pooling useful. In addition, the deeper dataset layers may allow better adaptation/harmonization in the representation to the characteristics of individual datasets. For posterity, we will also run this with the maximum amount of pre-training data to see how far we can push the boundaries of the downstream tasks.

---

### Note · Authors · 2025-01-22

I have read and agree with the venue's withdrawal policy on behalf of myself and my co-authors.